# The Genetic Structure of Cape Verdean Population Revealed by Y-Chromosome STRs

**DOI:** 10.3390/genes16090999

**Published:** 2025-08-25

**Authors:** Rita Costa, Jennifer Fadoni, António Amorim, Laura Cainé

**Affiliations:** 1Faculty of Medicine, Porto University, 4200-319 Porto, Portugal; up201906725@fc.up.pt; 2National Institute of Legal Medicine and Forensic Sciences, I.P., North Branch, 4050-202 Porto, Portugal; jennifer.n.fadoni@inmlcf.mj.pt (J.F.); antonio.j.amorim@inmlcf.mj.pt (A.A.); 3LAQV&REQUIMTE, Laboratory of Applied Chemistry, Department of Chemical Sciences, Faculty of Pharmacy, University of Porto, 4050-313 Porto, Portugal; 4National Institute of Legal Medicine and Forensic Sciences, I.P., Centre Branch, 3000-548 Coimbra, Portugal; 5Faculty of Sciences, Lisbon University, 1749-016 Lisboa, Portugal

**Keywords:** population genetics, haplotype, Y chromosome, Cape Verde, Argus Y28, forensic genetics

## Abstract

**Background/Objectives:** Y-chromosomal short tandem repeats (Y-STR) are genetic markers widely used in forensic and population genetics. However, despite their importance, many populations remain under-represented in published studies and genetic databases. One such population is the Cape Verdean, which, despite its unique history of admixture between European and sub-Saharan African populations, continues to be under-represented in global Y-STR reference databases. This study aims to characterize the Y-STR haplotype diversity and paternal lineage composition of the Cape Verdean population using a high-resolution STR panel. **Methods:** A total of 143 unrelated Cape Verdean men were analyzed using a set of 26 Y-STR loci, including rapidly mutating markers. Allele and haplotype frequencies were calculated, along with standard forensic parameters such as gene and haplotype diversity. Paternal lineages were inferred, and genetic relationships with other populations were evaluated using distance-based and graphical methods. **Results:** A total of 135 haplotypes were detected, with 88.8% being unique, yielding a haplotype diversity of 0.999. The most common haplogroups reflected both West African and European ancestry. Genetic distance analysis positioned the Cape Verdean population between African and European groups, supporting its intermediate and admixed genetic background. **Conclusions:** This study provides the first high-resolution Y-STR dataset for Cape Verdeans, contributing valuable reference data for forensic casework and population genetic studies. The results highlight the utility of extended Y-STR panels in admixed populations and underscore the need to enhance the representation of admixed populations in international forensic reference databases.

## 1. Introduction

Cape Verde is an archipelago located off the west coast of Africa in the Atlantic Ocean, discovered by Portuguese navigators in the 15th century. Permanent settlement began around 1460, driven by the arrival of a small number of European men, mainly Portuguese, and a much larger number of enslaved sub-Saharan Africans, primarily from the Senegambia region. Historical accounts also report the presence of Guanche individuals from the Canary Islands among the first slaves. This founding population laid the groundwork for the unique demographic and genetic landscape observed in the archipelago today [1,2,3,4,5] (Figure 1).

Over the centuries, Cape Verde has experienced intense migratory flows, mainly involving sub-Saharan Africans and Europeans. This admixture has led to an asymmetric genetic pattern, with a predominance of African maternal lineages and European paternal lineages, as documented by previous genetic studies [6,7].

Beyond its unique demographic history, the Cape Verdean population maintains a strong historical and cultural connection with Portugal, reflected in significant and sustained migration [8,9]. According to the 2023 Migration and Asylum Report, Cape Verdeans are the third-largest foreign community legally residing in Portugal, with 48,885 residents, accounting for 4.7% of the total foreign population. However, this figure does not fully capture the true demographic weight of Cape Verdeans due to the increasing acquisition of Portuguese nationality and ongoing migration to other countries [10,11].

These migration patterns and historical admixture have important implications in forensic genetics. Genetic studies indicate that the Cape Verdean population exhibits substructure and diverges significantly from other sub-Saharan African groups. Notably, short tandem repeat (STR) marker analysis has revealed differences between the northern (Barlavento) and southern (Sotavento) islands, likely due to genetic drift and limited gene flow. Such differentiation underscores the importance of establishing population-specific reference data, particularly for forensic applications [2,6].

Y-chromosomal STRs (Y-STRs) are commonly used in forensic casework involving male individuals, such as sexual assault investigations and paternity testing [12]. Among them, rapidly mutating Y-STRs (RM Y-STRs), which have elevated mutation rates (greater than 10^−2^), are especially valuable for distinguishing closely related men, as they provide higher resolution in pedigree analysis and in complex forensic cases [13,14].

The interpretation of Y-STR profiles relies on the availability of comprehensive population data. For this purpose, international databases like the Y Chromosome Haplotype Reference Database (YHRD) are essential tools that enable global comparison of haplotypes and support statistical evaluation of matches in forensic casework [15,16]. However, the Cape Verdean population remains under-represented in the YHRD, especially with respect to data generated from expanded multiplex kits such as the Argus Y-28, which includes both conventional and rapidly mutating loci, with only 117 haplotypes currently registered for 17 loci and none for 23 or more loci (https://yhrd.org (accessed on 20 August 2025)) [15,17,18,19].

Although previous studies have described the genetic characteristics of the Cape Verdean population, none have provided high-resolution data on allele and haplotype frequencies using expanded Y-STR panels, limiting their forensic applicability [2,6].

The aim of this study is to determine allele frequencies, haplotype diversity, and forensic parameters for 26 Y-STR loci in a sample of men residing in continental Portugal with Cape Verdean paternal ancestry, using an Argus Y-28 QS kit (QIAGEN, Hilden, Germany), and to contribute these data to the YHRD. This will enhance the representation of the Cape Verdean population in international forensic reference databases and support reliable Y-STR-based identification in medico-legal contexts.

## 2. Materials and Methods

### 2.1. Sample Collection

A total of 143 samples (135 dried blood spots and 8 buccal swabs) from unrelated men of Cape Verdean paternal ancestry were selected and analyzed. Paternal ancestry was assessed based on the individual’s identification documents, which confirmed their place of birth. Information regarding paternal lineage was further obtained from the individual’s self-reported account of their father’s origin. The samples, collected between 2012 and 2022 from paternity cases available at INMLCF, I.P., were fully anonymized prior to analysis, with no personal information of the individuals being accessible, except for their paternal ancestry. They remained stored until their use in this study (2024–2025). Buccal swabs were stored at room temperature in paper envelopes in a dry and dark environment until DNA extraction.

This study received ethical approval from the Ethics Committee of the National Institute of Legal Medicine and Forensic Sciences. The approval was granted on 25 July 2024, under the reference code CE-25/2024. The use of these samples complied with Portuguese legislation (Law No. 12/2005, of January 26) and with INMLCF regulations, which permit the use of anonymized samples stored for more than two years after casework completion.

### 2.2. DNA Extraction and Y-STR Fragment Analysis

The saliva samples collected using buccal swabs were extracted with a PrepFiler Express™ Forensic DNA Extraction Kit (Thermo Fisher Scientific, Waltham, MA, USA), following the manufacturer’s protocol. The blood samples on FTA^®^ cards were processed by direct amplification, without any prior DNA extraction procedure. Amplification of 26 Y-STR loci was performed using an Investigator^®^ Argus Y-28 QS PCR kit (QIAGEN, Hilden, Germany) on Applied Biosystems GeneAmp^®^ PCR System 9700 thermal cyclers (Thermo Fisher Scientific, Waltham, MA, USA). Reactions followed the manufacturer’s protocol, with two modifications to optimize reagent use and prevent overamplification: the reaction volume was halved to 12.5 μL, and the number of PCR cycles was reduced to 28.

For more details, the PCR reaction was performed using the following program: pre-denaturation at 96 °C for 12 min; denaturation at 96 °C for 10 s, annealing at 61.5 °C for 85 s, and extension at 72 °C for 5 s, for 28 cycles; and final extension at 68 °C for 5 min and 60 °C for 5 min, with maintenance at 10 °C. Control DNA 9948 (QIAGEN, Hilden, Germany) was used as the positive control and ddH*_2_*O as a negative control (without DNA) for each batch of Y-STR fragment analysis. The Investigator^®^ Argus Y-28 QS PCR kit included 20 standard Y-STR loci (DYS389I, DYS391, DYS389II, DYS533, DYS390, DYS458, DYS393, DYS19, DYS437, DYS460, YGATAH4, DYS448, DYS439, DYS549, DYS438, DYS456, DYS643, DYS635, DYS385, DYS392), 6 rapidly mutating markers (DYS449, DYS481, DYS570, DYS576, DYS518, DYS627), and quality sensors QS1 and QS2. PCR products were prepared for capillary electrophoresis (CE) by adding 1 μL of each PCR product to a mixture of 12 μL Hi-Di™ Formamide (Thermo Fisher Scientific, Waltham, MA, USA) and 0.5 μL of DNA Size Standard 24plex (BTO) (QIAGEN, Hilden, Germany). The reaction plate was then heat denatured at 95 °C for 3 min and subsequently cooled using a thermal cycler set to 4 °C. Capillary electrophoresis was carried out on an ABI 3500 Genetic Analyzer (Thermo Fisher Scientific, Waltham, MA, USA). DNA typing was carried out following the manufacturer’s protocol, using the provided locus panel, allele bins, and allele designations based on the supplied allelic ladder. The results were processed and analyzed using GeneMapper ID-X version 1.6 software (Thermo Fisher Scientific, Waltham, MA, USA), following the reference allelic ladder for genotype assignment. Reagent blanks were included alongside each batch of samples to verify the accuracy and reliability of the analyses. Additionally, samples that showed apparent null alleles, off-ladder (OL) peaks, or mutations in specific markers were reamplified to confirm the observed results.

### 2.3. Statistical Analyses

The number of distinct haplotypes, the frequency of unique haplotypes, discrimination capacity (DC), haplotype match probability (HMP), and haplotype diversity (HD) were calculated using a direct counting method in Microsoft Office Excel (version 16.94). The haplotype diversity (HD) was calculated applying the formula HD = n(1 − ∑pi^2^)/(n − 1), where n is the sample size, and pi is the frequency of the i-th haplotype [20]. The discrimination capacity (DC) was calculated as the ratio between the number of different haplotypes and the total number of individuals in the sample. The frequency of unique haplotypes was determined as the ratio between the number of unique haplotypes and the total number of individuals analyzed in the sample. The haplotype match probability (HMP) was calculated by summing the squared relative frequencies of all observed haplotypes (HMP = ∑pi^2^), where pi represents the frequency of the i-th haplotype. STRAF software (STR Analysis for Forensics) (version 2.2.2) was used to calculate forensic parameters, including allele frequencies at each locus, gene diversity (GD), polymorphism information content (PIC), and power of discrimination (PD) [21]. Pairwise genetic distances (R_ST_), estimated in Arlequin v3.5.2.2 software, were used to measure genetic distances between the samples from this study and the other compiled populations. For inter-population comparisons, the DYS389II allele length was calculated by subtracting the repeat count at DYS389I from that at DYS389II. For this analysis, only 17 Y-STR markers common to all population datasets were used. Using R statistical software version 4.0, a multidimensional scaling (MDS) plot was generated to visualize the relationships between populations based on the calculated matrix of pairwise genetic distances. Haplogroup prediction was performed using NevGen Y-DNA Haplogroup Predictor (https://www.nevgen.org, accessed on 16 June 2025) [22], assigning the haplogroup with the highest membership probability based on Y-STR haplotype.

## 3. Results

### 3.1. Analysis of Haplotype and Allele Diversity (and Forensic Parameters)

A total of 135 different haplotypes were obtained from 143 Cape Verdean samples, of which 127 (88.8%) were unique. The complete haplotype data for all 26 Y-STR loci are detailed in Appendix A. The allelic diversity observed across the loci ranged from 4 distinct alleles at DYS391 and DYS460 to 13 distinct alleles at DYS449, as shown in Appendix A. Allele frequency data for all loci are available in Appendix A. The haplotype match probability (HMP) was calculated to be 0.0078, and the haplotype diversity (HD) was found to be 0.999. The corresponding allelic frequencies varied from 0.0070 to 0.6643. The lowest GD value was 0.497 in locus DYS391 and the highest was 0.859 in locus DYS627. Among the male Cape Verdean samples, biallelic patterns were observed. These included the DYS389II locus (alleles 29/30 and 28/29), DYS437 (alleles 14/15), DYS448 (alleles 19/20, 20/21, and 19/22), and DYS439 (alleles 10/11), suggesting possible duplication events or structural variations on the Y chromosome. Additionally, microvariant alleles such as 17.2 and 16.2 were detected at the DYS458 locus, further highlighting the genetic diversity and mutational dynamics within the studied population. No meaningful differences in profile completeness were observed between saliva and blood samples. Nonetheless, blood stains occasionally exhibited slightly higher signal intensities and a slightly elevated level of background peaks. These variations did not affect the overall quality or the interpretation of the profiles.

### 3.2. Haplogroup Distribution

Out of the 143 Y-STR profiles analyzed, haplogroups were predicted for 120 individuals with a probability greater than 50% using the NevGen Y-DNA Haplogroup Predictor (https://www.nevgen.org, accessed on 16 June 2025) [22]. Within this subset, a total of 13 distinct haplogroups were identified: E1a, E1b1a, E1b1b, E3a, G2a2b1, I2a2a, I2a2b, J1a2a1a2, J2b2a, L1a, L1b, R1b, and T. The most frequently observed haplogroups were R1b and E1b1a (Table 1). Interestingly, haplogroups R and E showed similar overall frequencies in the sample, with R1b representing 44% and the combined E sub-haplogroups accounting for 45%. This suggests a balanced genetic contribution from both European (R) and African (E) paternal lineages. The remaining haplogroups appeared in lower frequencies, reflecting additional genetic diversity from Middle Eastern and Asian origins.

### 3.3. Genetic Distance Analysis Based on R_ST_ Distances

To explore the genetic relationships between the Cape Verdean population and other African and European groups, genetic distances were estimated based on the sum of squared size differences (R_ST_) between the haplotype distributions observed for 17 Y-STR loci. Comparative analysis could not be performed for the entire set of loci due to the absence of corresponding population data. However, for 17 loci in common (DYS389I, DYS391, DYS389II, DYS390, DYS458, DYS393, DYS19, DYS437, YGATAH4, DYS448, DYS439, DYS438, DYS456, DYS635, DYS392, DYS385a, DYS385b), comparisons were possible using published data from Guinea Bissau [23], Ghana [24], Nigeria [25], Cameroon [26], Angola [27], Spain [25], Portugal [28], and Morocco [29]. The pairwise R_ST_ values and corresponding *p*-values between the Cape Verdean population and the eight reference populations are displayed in Appendix A. The resulting distances were visualized through a multidimensional scaling (MDS) plot (Figure 2). The lowest genetic distances were observed with Cameroon (R_ST_ = 0.01230), Morocco (R_ST_ = 0.01747), Portugal (R_ST_ = 0.02098), and Angola (R_ST_ = 0.02100), indicating greater genetic affinity of Cape Verde with these African and Iberian populations. Although Spain showed a slightly negative RST value (–0.00347), this difference was not statistically significant (*p* = 0.88288). The highest genetic distances were recorded with Ghana (R_ST_ = 0.04314), Guinea-Bissau (R_ST_ = 0.03635), and Nigeria (R_ST_ = 0.03307). Among the pairwise comparisons, only some R_ST_ distances involving the Cape Verdean population were statistically significant (*p* < 0.05), as presented in Appendix A. The statistical analysis of R_ST_ distances revealed significant genetic differentiation between the Cape Verdean population and six of the eight populations compared. *p*-values below the conventional threshold of 0.05 were observed in the comparisons with Ghana (*p* = 0.00000), Nigeria (*p* = 0.02703), Cameroon (*p* = 0.01802), Angola (*p* = 0.01802), Portugal (*p* = 0.00000), and Morocco (*p* = 0.03604), indicating genetic divergence from these groups. In contrast, the comparisons with Guinea-Bissau (*p* = 0.0901) and Spain (*p* = 0.8829) did not reach statistical significance, suggesting no strong evidence of genetic differentiation from these populations. This may be consistent with a relatively closer genetic relationship, particularly with Iberian and some West African populations. The two-dimensional MDS plot provided an initial visualization of genetic relationships. In this representation, a cluster can be observed in which Cape Verde is positioned near Spain, Portugal, Ghana, and Guinea-Bissau, suggesting relative genetic proximity among these populations. However, the proportion of variance explained (PVE) was relatively low, with the first dimension accounting for 49.4% and the second for only 7.1%. These low cumulative values indicate that the 2D configuration does not accurately preserve the original genetic distances. Therefore, a three-dimensional MDS plot was generated (Appendix A) to enhance the spatial resolution and improve the interpretation of population structure. Together, these results highlight the importance of combining genetic distances with statistical support and appropriate dimensional representations when interpreting population affinities. This integrative approach contributes to a more accurate understanding of the paternal genetic structure and the intermediate position of the Cape Verdean population between African and European lineages.

## 4. Discussion

The genetic analysis of 143 unrelated Cape Verdean men using 26 Y-STR loci from an Argus Y-28 QS kit revealed a high level of haplotype diversity (HD = 0.999), with 135 different haplotypes and 88.8% being unique. These values reflect a high discrimination capacity, confirming the forensic utility of this multiplex system in admixed populations such as Cape Verde. The presence of only a few (*n* = 8) repeated haplotypes among unrelated individuals supports the system’s ability to differentiate between male individuals, including those with potential close kinship, particularly due to the inclusion of rapidly mutating Y-STRs that enhance resolution. The practical value of rapidly mutating Y-STRs lies in their elevated mutation rates, which generate novel allelic differences even between close paternal relatives, such as brothers or father–son pairs. This mutational dynamic reduces the probability of observing identical haplotypes among relatives and therefore increases the power of Y-STR analysis to exclude or distinguish individuals in forensic contexts. Such resolution is particularly important in cases involving sexual assault with multiple male contributors or in kinship testing scenarios where conventional Y-STR panels may yield indistinguishable profiles [14].

The detection of microvariant alleles and biallelic patterns in this study can be attributed to well-known mutational mechanisms affecting Y-STRs. Microvariants are most often generated by slippage events during DNA replication, where partial repeat units are inserted or deleted, producing intermediate allele sizes [13]. In contrast, duplicated alleles at certain loci may reflect structural rearrangements of the Y chromosome, such as segmental duplications or gene conversion events, which have been described in other populations [16,30]. Although relatively rare, these phenomena are relevant in forensic practice as they may influence allele designation and the interpretation of Y-STR profiles in casework.

From a forensic perspective, the availability of population-specific Y-STR reference data is essential for accurate statistical interpretation in casework. This need is particularly critical for admixed populations such as Cape Verdeans, whose paternal genetic profiles result from asymmetric historical admixture between European (predominantly R1b) and West African (mainly E1b1a and E1b1b) lineages [6,7]. In addition to the admixture events associated with the transatlantic slave trade, the observed genetic affinities with Iberian populations may also reflect more recent migratory flows during the 20th century, when thousands of Cape Verdeans settled in Portugal and other parts of Europe for economic and political reasons [10,11]. While the primary source of enslaved individuals brought to the archipelago during the transatlantic slave trade is historically associated with the Senegambian region [5], haplotypes from that area are under-represented or even absent in current Y-STR databases, such as YHRD [15,16]. This limits both comparative analyses and forensic resolution, especially when evaluating the statistical weight of evidence in investigations involving biogeographic ancestry inference. This absence of available reference data prevented the direct inclusion of populations from Senegal and Gambia in the RST-based comparisons, despite their well-documented historical relevance for the genetic makeup of Cape Verdeans. Additionally, many existing studies on African and admixed populations rely on panels with a reduced number of Y-STR loci or use datasets with limited haplotypic resolution, which restricts the robustness of comparative population analyses [31]. This limitation is particularly problematic for admixed populations or those with recent common paternal ancestors, where finer resolution is needed to distinguish close male relatives. In contrast, the use of 26 Y-STR loci in this study, including rapidly mutating markers, enabled the detection of rare allelic patterns such as microvariants and biallelic loci [13,14]. These markers are particularly informative in complex forensic scenarios, including sexual assault or kinship testing, where male individuals of the same paternal lineage must be distinguished. The genetic distinctiveness of Cape Verdeans also has practical implications in forensic casework. Y-STR profiles from individuals with Cape Verdean ancestry may be misclassified if interpreted against reference populations that do not adequately reflect their genetic background. This risk has been highlighted in studies emphasizing the forensic implications of misinterpretation in cases involving under-represented populations, particularly when Y-STR evidence is central to the investigation [32]. It should be noted that haplogroup prediction based on Y-STR profiles using NevGen is inherently probabilistic and particularly prone to misclassification in admixed populations such as Cape Verdeans. While SNP-based genotyping would provide more reliable assignments, the present results should be regarded as probabilistic inferences rather than definitive classifications. The RST-based comparisons performed in this study provide further insight into the genetic structure of the Cape Verdean population within a broader African–European context. The multidimensional scaling (MDS) analysis provided a visual approximation of genetic relationships among the studied populations. However, due to the limited amount of genetic variance represented in the two-dimensional configuration, the spatial distribution observed should be interpreted with caution. To mitigate this limitation, a three-dimensional MDS plot was generated, offering a more accurate depiction of the underlying genetic structure. In this higher-resolution model, the Cape Verdean population appears to occupy an intermediate position between Iberian and West/Central African groups. While this distribution does not reflect strict geographic proximity, it is consistent with the historical and demographic processes that shaped the paternal genetic landscape of Cape Verde, including founder effects, asymmetric admixture, and successive migratory events. Although this study focuses on Cape Verdean individuals residing in Portugal, the findings may not be directly generalizable to the population living in the Cape Verde islands due to possible migration effects, integration processes, and genetic drift within the diaspora. Moreover, the sample size (*n* = 143) reflects the limited availability of Cape Verdean paternal lineage samples accessible within the studied diaspora context, which inevitably constrains the statistical power of some analyses. These findings contribute valuable high-resolution data to the global Y-STR reference framework, particularly for under-represented admixed populations, and underscore the need to broaden the geographic and demographic scope of future studies [25,30,33]. Future studies should aim to include a broader sampling of individuals from specific islands, as well as from Cape Verdean communities in other regions of the diaspora, to build a more comprehensive and representative genetic dataset for forensic and population genetic purposes [8,9].

## 5. Conclusions

This study presents the first comprehensive analysis of 26 Y-STR loci in a sample of Cape Verdean men residing in Portugal, revealing high haplotype diversity and excellent discriminatory power. These findings underscore the effectiveness of the Argus Y-28 QS kit in forensic applications involving admixed populations. The detection of unique haplotypes, including rare allelic variants such as microvariants and biallelic patterns, highlights the added value of using extended Y-STR panels, especially those including rapidly mutating loci, for resolving cases involving close paternal relatives. Given the genetic distinctiveness and under-representation of Cape Verdean populations in international forensic databases, this dataset strengthens the basis for more reliable statistical evaluations in forensic casework and ancestry inference involving individuals of Cape Verdean descent. To ensure fair and accurate interpretation of Y-STR evidence, it is essential to expand population-specific reference databases, particularly for admixed and minority populations. Future research should prioritize the inclusion of samples from different Cape Verdean islands and other diaspora communities to refine our understanding of their genetic structure and improve global forensic practices. In addition, our findings highlight the importance of expanding forensic reference databases to include samples from diaspora communities. When island-based sampling is limited, diaspora populations can provide valuable insights into the genetic structure of under-represented groups, thereby strengthening the reliability of Y-STR data for both forensic and population genetic applications. Incorporating such data will ultimately contribute to more equitable forensic investigations and support judicial systems with scientifically robust genetic evidence.

## Figures and Tables

**Figure 1 genes-16-00999-f001:**
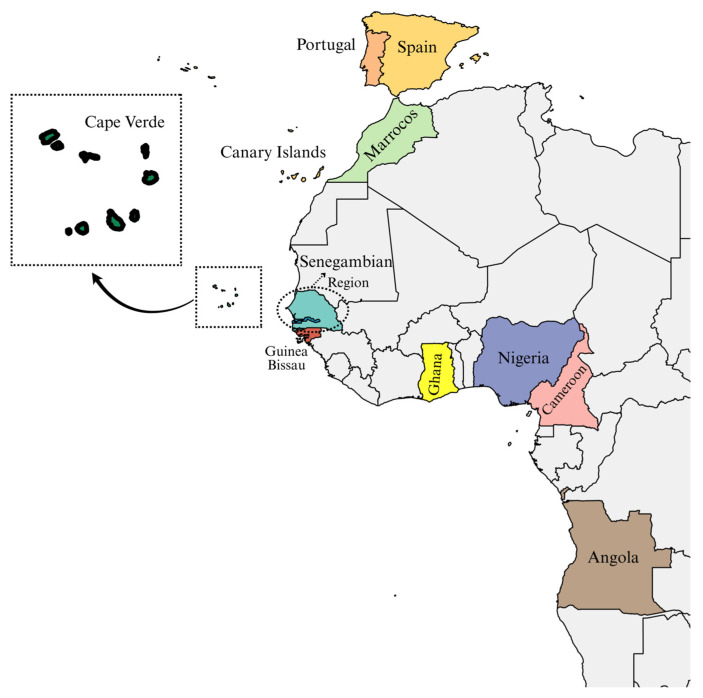
Geographic location of Cape Verde and the populations that contributed to the genetic landscape of the archipelago. Only countries with available Y-STR haplotype data are represented. The map was created using MapChart (https://www.mapchart.net, accessed on 2 July 2025).

**Figure 2 genes-16-00999-f002:**
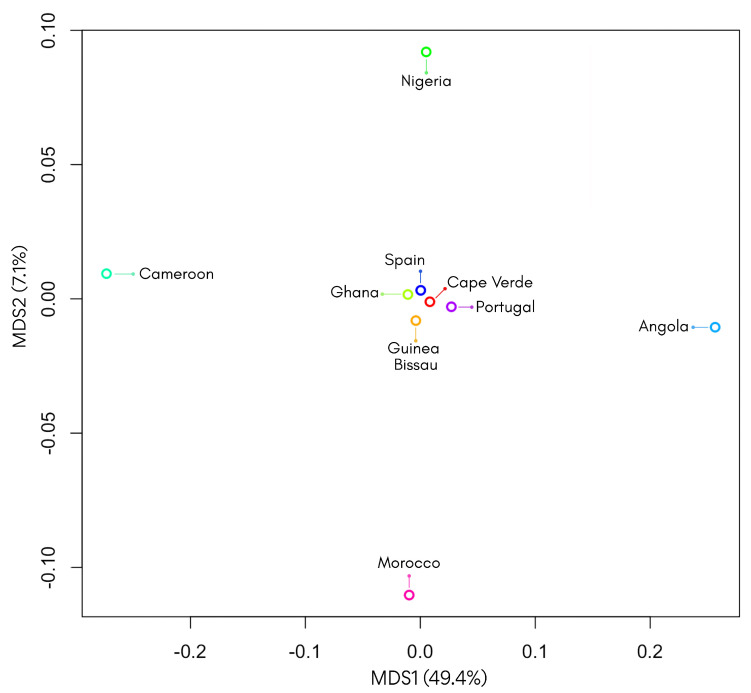
Multidimensional scaling (MDS) plot based on R_ST_ genetic distances among Cape Verde and reference populations. The analysis was performed using 17 shared Y-STR loci. The plot illustrates the genetic affinities between Cape Verde and other European and African populations, showing Cape Verde clustering closely with Guinea-Bissau, Spain, Ghana, and Portugal, while more distant populations include Angola, Cameroon, Nigeria, and Morocco. The percentages on the axes correspond to the proportion of variance explained (PVE) by each dimension.

**Table 1 genes-16-00999-t001:** Y-chromosome haplogroup and ancestry frequencies in Cape Verde individuals, with predicted haplogroups (*n* = 120).

Haplogroup (H)	Sub-Haplogroup (SH)	Frequency SH (%)	Frequency H (%)	Ancestry
E	E1a	3%	45%	African
	E1b1a	28%		
	E1b1b	14%		
	E3a	1%		
L	L1a	1%	2%	Asian
	L1b	1%		
I	I2a2a	1%	4%	European
	I2a2b	3%		
R	R1b	44%	44%	
G	G2a2b1	1%	1%	Middle Eastern
J	J1a2a1a2	2%	4%	
	J2b2a	3%		
T	-	1%	1%	

## Data Availability

The original contributions presented in this study are included in this article/Appendix A. Further inquiries can be directed to the corresponding author(s).

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
