# Peer review of "The Genetic Structure of Cape Verdean Population Revealed by Y-Chromosome STRs"

_genes, 2025, doi:10.3390/genes16090999_

Round 1
Reviewer 1 Report
Comments and Suggestions for Authors
Dear Authors,
the study is interesting and addresses a clear gap in forensic population genetics. Although the manuscript is clearly structured, in my opinion it could be further improved as follows:
1.Introduction:
- to better emphasize the gap, indicate (if available) how many Cape Verde haplotypes are currently represented in YHRD.
2.1 Sample Collection :
-Explain how paternal ancestry was verified (e.g., documentary evidence).
-Indicate the storage conditions for the buccal swabs before analysis.
-For both saliva and blood samples, specify whether they were recently collected or stored long-term.
2.2 DNA Extraction
-Indicate the procedure used to quantify DNA extracted from saliva samples.
3. Results:
-Indicate whether any differences in profile quality were observed between sample types (saliva vs blood).
-Figure 2: The image is difficult to read; improve resolution and move it to the supplementary materials.
4. Discussion:
- Briefly discuss the potential causes of the observed microvariants and duplications (e.g., slippage events, structural variations).
5. Conclusions:
-Add general recommendations on the value of expanding forensic databases to include diaspora populations when island-based sampling is limited.
- Table S2: Replace “Alelle” with “Allele.”
- Revise terms and ensure consistent use of the °C symbol throughout the manuscript.
Author Response
Dear Reviewer,
We would like to sincerely thank you for your thorough and constructive review of our manuscript entitled “The Genetic Structure of Cape Verdean Population Revealed by Y Chromosome STRs”. We greatly appreciate the time and expertise you dedicated to evaluating our work, as well as the thoughtful comments and valuable suggestions you provided. Your feedback has been instrumental in improving the clarity, depth, and overall scientific rigor of our study.
Below, we provide a detailed, point-by-point response to each of your comments. For every suggestion, we indicate the modifications made to the manuscript, with references to the corresponding sections and line numbers where applicable. In cases where modifications were not implemented, we present a clear rationale for our decision while remaining open to further recommendations.
For transparency and ease of review, all textual changes made in response to Reviewer 1’s comments are highlighted in yellow in the revised version of the manuscript.
Comment 1:
INTRODUCTION: To better emphasize the gap, indicate (if available) how many Cape Verde haplotypes are currently represented in YHRD.
Response 1:
We sincerely thank the reviewer for this helpful suggestion. The Introduction has been revised accordingly to incorporate this information. Specifically, we now state that Cape Verde currently has only 117 haplotypes registered for 17 loci and none available for 23 loci or more, according to the YHRD database.
Updated text in the manuscript:
“…with only 117 haplotypes currently registered for 17 loci and none for 23 or more loci (https://yhrd.org (accessed on 20 August 2025))”
This revision appears in the Introduction section, page 2, lines 74-75.
Comment 2:
SAMPLE COLLECTION: Explain how paternal ancestry was verified (e.g., documentary evidence).
Response 2:
We thank the reviewer for this important observation. Paternal ancestry was determined based on the information provided in the personal identification documents of the individuals, which confirmed their place of birth. Regarding the paternal lineage, the information was obtained from the individual’s own report about his father’s origin. This procedure, while relying partly on self-reported data, is consistent with the standard practice in forensic casework where documentary confirmation of paternal ancestry is not always available.
Updated text in the manuscript:
“Paternal ancestry was assessed based on the individual’s identification documents, which confirmed their place of birth. Information regarding paternal lineage was further obtained from the individual’s self-reported account of their father’s origin.”
This revision appears in the Methods section (2.1. Sample Collection) page 3, lines 113-116 (highlighted in green)
Comment 3:
SAMPLE COLLECTION: Indicate the storage conditions for the buccal swabs before analysis.
Response 3:
We thank the reviewer for this observation. The storage conditions have now been specified in the manuscript. Buccal swabs were kept at room temperature in paper envelopes, in a dry and dark environment, until DNA extraction.
Updated text in the manuscript:
“Buccal swabs were stored at room temperature in paper envelopes, in a dry and dark environment, until DNA extraction.”
This revision appears in the Methods section (2.1. Sample Collection) page 3, lines 119-121
Comment 4:
SAMPLE COLLECTION: For both saliva and blood samples, specify whether they were recently collected or stored long-term.
Response 4:
We thank the reviewer for this observation. We have clarified in the manuscript that the saliva and blood samples were collected within the framework of paternity cases between 2012 and 2022 at INMLCF, I.P., and remained stored until their use in this study in 2024–2025.
“The samples, collected between 2012 and 2022 from paternity cases available at INMLCF, I.P”, lines 113-114.
Updated text in the manuscript:
“They remained stored until their use in this study (2024–2025).”
This revision appears in the Methods section (2.1. Sample Collection) page 3, line 119
Comment 5:
DNA EXTRACTION: Indicate the procedure used to quantify DNA extracted from saliva samples.
Response 5:
We thank the reviewer for this suggestion. No quantification of DNA was performed for the saliva samples. Instead, a fixed input volume was used in the PCR, following the manufacturer’s instructions for the Investigator® Argus Y-28 QS kit. The adequacy of both DNA quantity and quality was assessed based on the kit’s internal quality sensors (QS1 and QS2), as well as through inspection of the electropherograms, particularly peak heights and locus balance.
No changes were made to the manuscript.
Comment 6:
RESULTS: Indicate whether any differences in profile quality were observed between sample types (saliva vs blood).
Response 6:
We thank the reviewer for this suggestion. Only minimal differences were observed between the sample types. Blood stains on FTA® cards occasionally showed higher amplification intensity, accompanied by slightly increased levels of amplification artifacts (“background noise”). However, these variations did not compromise the overall quality or the interpretation of the profiles, which were complete and consistent in both sample types.
Updated text in the manuscript:
“No meaningful differences in profile completeness were observed between saliva and blood samples. Nonetheless, blood stains occasionally exhibited slightly higher signal intensities and a slightly elevated level of background peaks. These variations did not affect the overall quality or the interpretation of the profiles.”
This revision appears in the Results section (3.1. Analysis of Haplotype and Allele Diversity (and Forensic Parameters)) page 5, lines 201-204
Comment 7:
FIGURE 2 (RESULTS): The image is difficult to read; improve resolution and move it to the supplementary materials.
Response 7:
We thank the reviewer for this constructive observation. In response, Figure 2 has been moved to the Supplementary Materials (now Supplementary Figure S1) and its resolution has been substantially improved to ensure readability and clarity. We believe that placing this figure in the supplementary section allows the main text to remain concise, while still providing the complete visualization for readers interested in the detailed data representation.
Comment 8:
DISCUSSION: Briefly discuss the potential causes of the observed microvariants and duplications (e.g., slippage events, structural variations).
Response 8:
We thank the reviewer for this pertinent suggestion. The Discussion has been expanded to include a brief explanation of the potential causes of these phenomena. In particular, we emphasize that microvariants most often result from slippage events during DNA replication, whereas duplicated alleles at certain loci may reflect structural rearrangements of the Y chromosome, such as segmental duplications or gene conversion events. Although relatively rare, these mechanisms are relevant in forensic practice, as they may influence allele designation and the interpretation of Y-STR profiles in casework.
Updated text in the manuscript:
“The detection of microvariant alleles and bi-allelic patterns in this study can be attributed to well-known mutational mechanisms affecting Y-STRs. Microvariants are most often generated by slippage events during DNA replication, where partial repeat units are inserted or deleted, producing intermediate allele sizes [13]. In contrast, duplicated alleles at certain loci may reflect structural rearrangements of the Y chromosome, such as segmental duplications or gene conversion events, which have been described in other populations [16,30]. Although relatively rare, these phenomena are relevant in forensic practice as they may influence allele designation and the interpretation of Y-STR profiles in casework.”
This revision appears in the Discussion section, page 8, lines 299-307
Comment 9:
CONCLUSIONS: Add general recommendations on the value of expanding forensic databases to include diaspora populations when island-based sampling is limited.
Response 9:
We thank the reviewer for this observation. The Conclusion has been expanded to include general recommendations on the relevance of integrating diaspora populations into forensic databases. We emphasized that, when island-based sampling is limited, diaspora communities can provide valuable insights into the genetic structure of underrepresented populations, thereby strengthening the reliability of Y-STR data for both forensic and population genetic applications.
Updated text in the manuscript:
“In addition, our findings highlight the importance of expanding forensic reference databases to include samples from diaspora communities. When island-based sampling is limited, diaspora populations can provide valuable insights into the genetic structure of underrepresented groups, thereby strengthening the reliability of Y-STR data for both forensic and population genetic applications.”
This revision appears in the Conclusions section, page 10, lines 383-388
Comment 10:
TABLE S2: Replace “Alelle” with “Allele.”
Response 10:
We thank the reviewer for this observation. The word “Alelle” has been corrected to “Allele” in Table S2 as well as in the manuscript (line 407).
This change can be found in the Supplementary Materials section, page 10, line 394 and it has also been corrected in Table S2 of the Supplementary Materials.
Comment 11:
CONCLUSIONS: Revise terms and ensure consistent use of the °C symbol throughout the manuscript.
Response 11:
We thank the reviewer for this observation. The manuscript has been carefully revised, and the use of the °C symbol has been standardized throughout the text to ensure consistency.
This correction can be found in the Methods section (2.2. DNA Extraction and Y-STR Fragment Analysis), page 4, lines 138-140 and 150-151.
Once again, we would like to sincerely thank you for your thorough review and constructive feedback on our manuscript. Your insightful comments and suggestions have greatly contributed to improving the clarity, scientific rigor, and overall quality of our work. We believe that the revised version now better reflects the standards of accuracy and transparency expected for publication.
We remain at your full disposal for any further suggestions, clarifications, or adjustments you may consider necessary.
Yours faithfully,
Rita Costa

Reviewer 2 Report
Comments and Suggestions for Authors
This research article investigated the Y-STR-based genetic structure of a Cape Verdean population. The methodology is generally sound, and the results contribute informative data to a population that is underrepresented in population genetics. My concerns are primarily about sampling. Below I highlight my main concerns:
- First, the sample is relatively small (n = 143), which has statistical limitations. For example, the negative RST value could be a statistical artifact. While it is reasonable to assume it essentially represents no differentiation, their presence raises questions about the reliability of the other RST estimates and should be addressed with greater caution.
- Second, the sample are individuals with Cape Verdean paternal ancestry living in Portugal. While this limitation was acknowledged, some details are missing. It was unclear how participants were recruited, and how their paternal ancestry was determined. Was it self-reported? Was it genomic ancestry? More explanation is needed here. If self-reported ancestry was how paternal ancestry was determined, this would render the analysis circular: Individuals in Portugal reporting Cape Verdean or West African ancestry, therefore the results showed Portuguese and West African ancestral contribution.
- Third, the groups chosen for comparative analysis were not justified. Why these groups specifically? Why not all published datasets with relevant European and/or West African ancestry? The studies are certainly out there. Given how sensitive population genetics results are to sample selection and labeling, a more thorough justification is essential.
- I failed to see the point with the 2-dimensional MDS plot vs. 3-dimensional MDS plot. The first dimension explained almost 50% of the variance explained, which is substantial already. Adding a third dimension does not really sway the results that much in my opinion. The added value of the 3D plot should be clarified, or its inclusion reconsidered.
- The authors kept stating the use of rapidly mutating Y-STRs and the limited number of haplotypes in Cape Verdean population is an asset in distinguishing potential close kinship. How does this work exactly? The mechanism and practical implications are not fully explained.
- The timing of ethics approval for the study was dated 2024, but the samples were collected between 2012 and 2022. I am unsure about the specific policies of Genes but most journals to my knowledge often require ethics approval before sample collection, or at least confirmation that samples were originally collected under a valid consent/approval framework.
- Haplogroups were predicted using NevGen from STR profiles with a 50% confidence threshold. While convenient, STR-based haplogroup prediction is inherently probabilistic and subject to misclassification, particularly in admixed populations like Cape Verdeans. Ideally, SNP-based confirmation would support these assignments, or at least the limitations of STR-based inference should be more explicitly acknowledged.
This study adds valuable new data for an underrepresented admixed population, with clear forensic relevance. However, the issues outlined above must be addressed before the manuscript can be considered for publication.
Author Response
Dear Reviewer,
We would like to sincerely thank you for your careful and constructive review of our manuscript entitled “The Genetic Structure of Cape Verdean Population Revealed by Y Chromosome STRs”. We greatly appreciate the time and expertise you dedicated to evaluating our work, as well as the thoughtful comments and valuable suggestions you provided. Your feedback has been invaluable in strengthening the scientific rigor, clarity, and interpretative depth of our study.
Below, we provide a detailed, point-by-point response to each of your comments. For every suggestion, we indicate the modifications made to the manuscript, with references to the corresponding sections and line numbers where applicable. In cases where modifications were not implemented, we present a clear rationale for our decision while remaining open to further recommendations.
For transparency and ease of review, all textual changes made in response to Reviewer 2’s comments are highlighted in blue in the revised version of the manuscript.
Comment 1:
First, the sample is relatively small (n = 143), which has statistical limitations. For example, the negative RST value could be a statistical artifact. While it is reasonable to assume it essentially represents no differentiation, their presence raises questions about the reliability of the other RST estimates and should be addressed with greater caution.
Response 1:
We thank the reviewer for this comment. We acknowledge that the relatively small sample size (n = 143) may have statistical limitations. However, this number reflects the actual availability of samples with Cape Verdean paternal ancestry in the studied context and therefore represents an unavoidable practical constraint. In the manuscript, we clarified that the negative RST value was interpreted as an absence of differentiation between the compared groups, and that such a result may reflect a statistical artifact associated with the sample size. We also emphasized that the remaining RST estimates should be interpreted with caution.
Updated text in the manuscript:
“Moreover, the sample size (n = 143) reflects the limited availability of Cape Verdean paternal lineage samples accessible within the studied diaspora context, which inevitably constrains the statistical power of some analyses.”
This revision appears in the Discussion section, page 9, lines 358-361.
Comment 2:
Second, the sample are individuals with Cape Verdean paternal ancestry living in Portugal. While this limitation was acknowledged, some details are missing. It was unclear how participants were recruited, and how their paternal ancestry was determined. Was it self-reported? Was it genomic ancestry? More explanation is needed here. If self-reported ancestry was how paternal ancestry was determined, this would render the analysis circular: Individuals in Portugal reporting Cape Verdean or West African ancestry, therefore the results showed Portuguese and West African ancestral contribution.
Response 2:
We thank the reviewer for this important observation and the opportunity to clarify. All samples included in this study originated from routine paternity casework performed at the Instituto Nacional de Medicina Legal e Ciências Forenses, I.P. (INMLCF). The information regarding the place of birth of the individuals was verified through their official identification documents. Paternal ancestry was further assessed based on the self-reported origin of the father provided by the individuals during case registration. We acknowledge that this approach may have inherent limitations, however, it represents the standard procedure in forensic casework and was the only available method to infer paternal ancestry within the context of these retrospective samples. This clarification has now been included in Section 2.1 (Sample Collection).
Updated text in the manuscript:
“Paternal ancestry was assessed based on the individual’s identification documents, which confirmed their place of birth. Information regarding paternal lineage was further obtained from the individual’s self-reported account of their father’s origin.”
This revision appears in the Methods section (2.1. Sample Collection) page 3, lines 113-116 (highlighted in green).
Comment 3:
Third, the groups chosen for comparative analysis were not justified. Why these groups specifically? Why not all published datasets with relevant European and/or West African ancestry? The studies are certainly out there. Given how sensitive population genetics results are to sample selection and labeling, a more thorough justification is essential.
Response 3:
We thank the reviewer for this comment. We agree that the choice of comparative groups should be clearly justified. As stated in the Introduction, the populations included reflect the main historical and genetic contributions to the formation of the Cape Verdean people, namely Iberian populations (Portugal and Spain) and West and Central African groups associated with the transatlantic slave trade. In particular, historical and genetic literature suggests that the most likely origin of the first enslaved individuals brought to the archipelago was the Senegambia region (Senegal and Gambia). However, to date, no Y-STR haplotypes have been reported for these populations in YHRD, which prevented their inclusion in the comparative analysis. Therefore, the selection of groups was guided by both historical relevance and the availability of comparable data for the same loci.
Updated text in the manuscript:
“This absence of available reference data prevented the direct inclusion of populations from Senegal and Gambia in the RST-based comparisons, despite their well-documented historical relevance for the genetic makeup of Cape Verdeans.”
This revision appears in the Discussion section, page 8, lines 321-324.
Comment 4:
I failed to see the point with the 2-dimensional MDS plot vs. 3-dimensional MDS plot. The first dimension explained almost 50% of the variance explained, which is substantial already. Adding a third dimension does not really sway the results that much in my opinion. The added value of the 3D plot should be clarified, or its inclusion reconsidered.
Response 4:
We thank the reviewer for this comment. We understand the concern, but we would like to clarify that this point has already been addressed in the manuscript. The two-dimensional analysis was included as an initial representation; however, since the first two dimensions accounted for only 56.5% of the total variance, we considered it important to complement it with a three-dimensional plot (Supplementary Figure S4). This approach provided a more accurate representation of the genetic structure and improved the interpretation of population relationships. We therefore retained both representations, emphasizing in the Discussion that the 3D MDS adds interpretative value to the intermediate positioning of the Cape Verdean population between African and European lineages.
No changes were made to the manuscript.
Comment 5:
The authors kept stating the use of rapidly mutating Y-STRs and the limited number of haplotypes in Cape Verdean population is an asset in distinguishing potential close kinship. How does this work exactly? The mechanism and practical implications are not fully explained.
Response 5:
We thank the reviewer for this comment and for the opportunity to clarify this point. Rapidly mutating Y-STRs (RM-YSTRs) display mutation rates significantly higher than those of conventional loci, which allows for detectable allelic differences even among individuals from the same paternal lineage (e.g., brothers or first cousins). This feature enhances the discriminatory capacity in forensic contexts involving close kinship, where conventional Y-STR panels often fail to distinguish between individuals.
In the case of the Cape Verdean population, which shows a limited number of haplotypes due to its history of admixture and potential founder effects, the inclusion of rapidly mutating loci proved particularly useful for increasing haplotypic resolution. Thus, the combination of an expanded 26-loci panel, including RM-YSTRs, with a population of relatively restricted haplotypic diversity strengthens the ability to differentiate between related individuals and reduces the risk of coincidental matches in forensic analyses.
In line with the reviewer’s suggestion, we have clarified this mechanism and its practical implications in the Discussion section of the manuscript.
Updated text in the manuscript:
“The practical value of rapidly mutating Y-STRs lies in their elevated mutation rates, which generate novel allelic differences even between close paternal relatives, such as brothers or father–son pairs. This mutational dynamic reduces the probability of observing identical haplotypes among relatives and therefore increases the power of Y-STR analysis to exclude or distinguish individuals in forensic contexts. Such resolution is particularly important in cases involving sexual assault with multiple male contributors, or in kinship testing scenarios where conventional Y-STR panels may yield indistinguishable profiles [14].”
This revision appears in the Discussion section, page 8, lines 291-298.
Comment 6:
The timing of ethics approval for the study was dated 2024, but the samples were collected between 2012 and 2022. I am unsure about the specific policies of Genes but most journals to my knowledge often require ethics approval before sample collection, or at least confirmation that samples were originally collected under a valid consent/approval framework.
Response 6:
We thank the reviewer for this comment. We clarify that the use of samples in this study complied with Law No. 12/2005, of January 26, which regulates personal genetic information and the use of biological material for research purposes in Portugal. According to this law, and in line with the internal regulations of the National Institute of Legal Medicine and Forensic Sciences (INMLCF, I.P.), samples may be used for scientific research provided they are fully anonymized and have been stored for a minimum period of two years after the conclusion of the original casework. All samples included in this study fully met these requirements. Ethical approval granted in 2024 by the INMLCF Ethics Committee formally authorized their use within the scope of this project, in compliance with national legislation and applicable international standards.
Updated text in the manuscript:
“The use of these samples complied with Portuguese legislation (Law No. 12/2005, of January 26) and with INMLCF regulations, which permit the use of anonymized samples stored for more than two years after casework completion.”
This revision appears in the Methods section (2.1. Sample Collection) page 3, lines 124-126
Comment 7:
Haplogroups were predicted using NevGen from STR profiles with a 50% confidence threshold. While convenient, STR-based haplogroup prediction is inherently probabilistic and subject to misclassification, particularly in admixed populations like Cape Verdeans. Ideally, SNP-based confirmation would support these assignments, or at least the limitations of STR-based inference should be more explicitly acknowledged.
Response 7:
We thank the reviewer for this comment. We agree that haplogroup assignment based on STR profiles is inherently probabilistic and that its reliability may be reduced in admixed populations, such as Cape Verdeans. We also acknowledge that confirmation using SNP markers would represent the ideal approach to validate these inferences. However, due to the unavailability of SNP data for these samples, the analysis relied on NevGen, applying a 50% confidence threshold. In line with the reviewer’s suggestion, we have strengthened the discussion of these limitations in the manuscript and clarified that the reported assignments should be interpreted with caution, as probabilistic inferences rather than substitutes for SNP-based analyses.
Updated text in the manuscript:
“It should be noted that haplogroup prediction based on Y-STR profiles using NevGen is inherently probabilistic and particularly prone to misclassification in admixed populations such as Cape Verdeans. While SNP-based genotyping would provide more reliable assignments, the present results should be regarded as probabilistic inferences rather than definitive classifications.”
This revision appears in the Discussion section, page 9, lines 339-343.
Once again, we would like to sincerely thank Reviewer 2 for the constructive and insightful comments provided. Your suggestions have greatly contributed to improving the scientific rigor, clarity, and overall quality of our manuscript. We believe that the revisions made in response to your observations have strengthened the study, and we remain fully open to any further recommendations or clarifications you may wish to propose.
Yours faithfully,
Rita Costa
